# Recall Distortion in Neural Network Pruning and the Undecayed Pruning Algorithm

**Aidan Good**[*]
Bucknell University

**Jiaqi Lin**[*]
Bucknell University

**Xin Yu**[*]
University of Utah

**Hannah Sieg**
Bucknell University

**Mikey Ferguson**
Bucknell University

**Shandian Zhe**
University of Utah

**Jerzy Wieczorek**
Colby College

**Thiago Serra**[†]
Bucknell University

## Abstract

Pruning techniques have been successfully used in neural networks to trade accuracy for sparsity. However, the impact of network pruning is not uniform: prior work has shown that the recall for underrepresented classes in a dataset may be more negatively affected. In this work, we study such relative distortions in recall by hypothesizing an intensification effect that is inherent to the model. Namely, that pruning makes recall relatively worse for a class with recall below accuracy and, conversely, that it makes recall relatively better for a class with recall above accuracy. In addition, we propose a new pruning algorithm aimed at attenuating such effect. Through statistical analysis, we have observed that intensification is less severe with our algorithm but nevertheless more pronounced with relatively more difficult tasks, less complex models, and higher pruning ratios. More surprisingly, we conversely observe a de-intensification effect with lower pruning ratios, which indicates that moderate pruning may have a corrective effect to such distortions.

## 1 Introduction

Back in the old days, network pruning was important to reduce the size of neural networks [23, 53, 33, 41, 24, 25]. With new advances being increasingly reliant on overwhelmingly large and costly models [3, 64, 10], network pruning is back to the game despite the gains in computing power.

Network pruning reduces the complexity of large models by setting a substantial number of parameters to zero and then using gradient descent to fine-tune the sparser model. There is typically a trade-off between making the model sparser—i.e., having fewer parameters, since those set to zero can simply be discarded—and keeping the model as accurate as possible [9], although moderate gains in sparsity may be obtained with little impact to accuracy [28]. However, recent studies have found that the side effects of network pruning on model performance are not evenly distributed [30, 55, 31, 1, 63, 69].

Those studies have considered the impact of network pruning on recall—i.e., the number of correct predictions per class—based on dataset representability. Namely, they have found that the degradation in performance is influenced by the uneven representation of classes [30, 55, 69] and features [31] as well as class complexity [55]. Hence, they corroborate long-standing concerns that unbalanced datasets yield models that are less accurate in and potentially harmful to minoritized groups [11, 7].

In this work, we complement those prior studies by posing recall distortion as inherent to network pruning even when the networks are trained and fine-tuned on datasets that are seemingly balanced; hence investigating when and how such distortions manifest, as well as how to reduce their effect.

---

[*]Equal contribution
[†]Corresponding author: `thiago.serra@bucknell.edu`

36th Conference on Neural Information Processing Systems (NeurIPS 2022).

More specifically, this paper presents the following contributions:

(i) We conduct a statistical study of model-level distortion to investigate how recall is affected by network pruning due to pruning ratios, dataset and model complexity, and pruning algorithms.

(ii) We observe an intensification effect, meaning that for classes with recall below accuracy we observe the relative difference between recall and accuracy negatively widening if the network is sufficiently pruned. For classes with recall above accuracy, we conversely observe that gap positively widening. The intensification correlates with excessive pruning ratios as well as more complex data and models, and is more pronounced with some pruning algorithms.

(iii) More surprisingly, we observe that otherwise network pruning has a corrective effect on recall differences, hence implying a de-intensification effect under moderate pruning.

(iv) We introduce a new gradient-based pruning method for networks trained with weight decay, Undecayed Pruning, which attenuates the intensification effect observed with other methods.

## 2    Related work

The use of large models has been justified and further encouraged by findings that the overparameterized regime may avoid the classic bias–variance trade-off [83, 6] and lead to better convergence during training [44, 66]. However, these models come with steep environmental footprint and hardware needs [64]. Not surprisingly, they represent a relevant application of network pruning [20, 78].

Network pruning has been motivated by parameter redundancy in models [13] and found to improve generalization [4] and robustness against adversarial manipulation [74, 35]. The amount of pruning that is tolerable by an architecture may depend on the task in which it is trained [46]. As observed on a recent survey [9], most approaches to network pruning are guided by either (i) removing the parameters with smallest absolute value [23, 53, 33, 22, 21, 43, 17, 15, 20, 68, 48]; or (ii) removing the parameters with smallest impact on the model [41, 24, 25, 39, 52, 14, 79, 82, 5, 42, 71, 47, 72, 76, 61, 80]. We may regard exact pruning as a special case of (ii) in which the model is not affected [59, 62, 60, 18]. Alternative approaches to network pruning include quantization [32, 50, 2], knowledge distillation [37, 45, 84, 73, 70, 67, 34, 77], and the use of regularization during training [81, 16, 56].

In this work, we contribute to the study of recall distortion due to model compression [30, 55, 31, 1, 63], which a particular focus on network pruning. Concurrent to our work, Tran et al. [69] framed these recall distortions in unbalanced datasets as a Matthew effect that can be attributed to model (Hessian loss and gradient flows) as well as data characteristics (input norms and distance to decision boundary). Pruning distortion has also been studied under other metrics, such as by focusing on the samples for which the classification is affected [30, 31, 36, 78] and the relationship between false positive and negative rates [8]. The issue is also studied in the context of other compression techniques such as knowledge distillation [27], in which some approaches to remedy this issue are focused on adjusting the loss function used in the compressed model [36, 78].

Such disproportional impact on recall across classes relates to fairness in machine learning. The lack of representativeness and the biased context in which data is collected as well as the lack of transparency may lead to models making life-altering decisions that negatively affect minoritized groups, such as in criminal justice [54, 11, 57]. These concerns have motivated an extensive discussion on fairness in machine learning [12] and on the proper assessment of datasets [19], models [51], and the circumstances in which they are applied [58]. In this work, we focus on studying the algorithmic bias [49, 29] of network pruning, that is, the bias that is either (i) not present in the input data nor the original model; or (ii) deteriorated or relieved by pruning. By subscribing to this line of study, our work aims at reducing the potential negative societal impacts associated with network pruning.

## 3    Pruning algorithms

We propose Undecayed Pruning (UP) by considering the interplay between the classic representatives of pruning methods described in Section 2: Magnitude Pruning (MP) and Gradient Pruning (GP).

**Magnitude Pruning**  MP is a simple but rather (mysteriously) effective technique of selecting among the parameters $\theta = \bar{\theta}$ the one with smallest absolute value to be pruned next from the network:

$$i = \arg\min_i \left\{ |\bar{\theta}_i| \right\}$$

A parameter having a smaller value may not necessarily imply lesser importance to the model [24] (hence the mystery part). However, it is worth noticing that this technique is usually applied to networks that were trained with regularization. Since we cannot add $L0$ regularization to the loss function for directly minimizing the number of nonzero parameters as this would make the loss function not differentiable for gradient descent, we can use $L1$ or $L2$ as a proxy, which induces the values of the parameters to be as small as possible. Although no parameter ends up being zero, some parameters become so small that making them equal to zero implies an almost negligible change.

**Gradient Pruning**  GP approximates the impact of modifying the parameters of a neural network by estimating the first-order variation of the loss function $\mathcal{L}$ on the training set between the trained parameters $\theta = \bar{\theta}$ to a new set of parameters $\theta$ through the Taylor series around $\theta = \bar{\theta}$:

$$\mathcal{L}(\theta) = \mathcal{L}(\bar{\theta}) + (\theta - \bar{\theta})\nabla\mathcal{L}(\bar{\theta}) + O(\|\theta - \bar{\theta}\|^2)$$

By assuming that $O(\|\theta - \bar{\theta}\|^2) \approx 0$ for small changes, we estimate the change to the loss function by pruning a single parameter $i$—i.e., with $\tilde{\theta}$ such that $\tilde{\theta}_i = 0$ and $\tilde{\theta}_j = \bar{\theta}_j$ for $j \neq i$—as follows:

$$\mathcal{L}(\tilde{\theta}) - \mathcal{L}(\bar{\theta}) \approx (\tilde{\theta} - \bar{\theta})\nabla\mathcal{L}(\bar{\theta}) = -\bar{\theta}_i\nabla_i\mathcal{L}(\bar{\theta})$$

Given the approximate nature of estimating the impact on the training set, which may not necessarily reflect on the test set, small absolute perturbations are often preferred to negative but larger variations. Hence, the choice of the parameter $i$ to be pruned next is effectively framed as follows:

$$i = \arg\min_i \left\{ |-\bar{\theta}_i\nabla_i\mathcal{L}(\bar{\theta})| \right\}$$

**Undecayed Pruning**  We aim to address what we believe is a double-edged approach to reducing the number of parameters. Namely, that it is common to use a term in the loss function to make the weights as small as possible, such as weight decay, and then that we use the gradient of the loss function for pruning weights while ignoring that the gradient is also affected by the weight decay term. We understand that simultaneously using regularization and GP is conflicting because sparsity is induced in two different ways. However, whereas GP has a more direct proxy to model impact, using some amount of regularization tends to be beneficial during training. Hence, we approach this by isolating the effect of regularization from the variation of the loss function. Due to its greater popularity, we consider $L2$ regularization–i.e,, weight decay–in what follows. Let us denote by $\mathcal{T}$ the loss function of a model after deducting weight decay with hyperparameter $\varepsilon$:

$$\mathcal{T}(\theta) = \mathcal{L}(\theta) - \frac{\varepsilon}{2}\|\theta\|^2$$

With $\nabla_i\mathcal{T}(\theta) = \nabla_i\mathcal{L}(\theta) - \varepsilon\theta_i$, we estimate the change to the alternative loss function $\mathcal{T}$ by pruning a single parameter $i$—i.e., with $\tilde{\theta}$ such that $\tilde{\theta}_i = 0$ and $\tilde{\theta}_j = \bar{\theta}_j$ for $j \neq i$—as follows:

$$\mathcal{T}(\tilde{\theta}) - \mathcal{T}(\bar{\theta}) \approx (\tilde{\theta} - \bar{\theta})\nabla\mathcal{T}(\bar{\theta}) = -\bar{\theta}_i\nabla_i\mathcal{T}(\bar{\theta}) = -\bar{\theta}_i\nabla_i\mathcal{T}(\bar{\theta}) = -\bar{\theta}_i\nabla_i\mathcal{L}(\bar{\theta}) + \varepsilon\bar{\theta}_i^2$$

In other words, discounting weight decay is equivalent to a specific balance of the criteria for MP and GP: the first term is equivalent to GP; the second term is curiously equivalent to MP; and the latter is prioritized in proportion to the weight decay hyperparameter $\varepsilon$. Moreover, if the neural network training converges to a local optimum, in which case $\nabla\mathcal{L}(\bar{\theta}) = 0$, then UP is equivalent to MP. Similar to the case of GP, we choose the parameter $i$ to prune based on the absolute impact on $\mathcal{T}$:

$$i = \arg\min_i \left\{ |-\bar{\theta}_i\nabla\mathcal{L}(\bar{\theta}) + \varepsilon\bar{\theta}_i^2| \right\}$$

The way that weights are now ranked reflects a weighted combination of the criteria used for MP and GP. This is particularly interesting because UP coincides with MP when the gradient is sufficiently close to zero, hence providing a principled argument for the effectiveness of MP under weight decay.

# 4   Model properties

Let $A(m)$ be the accuracy of an unpruned model $m$ as measured on the test data. In other words, $A(m)$ is the number of correct predictions divided by the number of samples. For a *pruning ratio* $t$, meaning that $t$ is the number of parameters before pruning divided by the number of parameters after pruning, let $A_t(m)$ denote the accuracy of model $m$ on the test data after pruning. For simplicity, we may assume that $t = 1$ if $t$ is absent from the notation and omit $m$ if always referring to the same model, and thus we may assume that $A = A(m) = A_1(m)$. The same applies to other metrics.

Similarly, let $R^c(m)$ be the *recall* for class $c$ of an unpruned model $m$ on the test data. In other words, $R^c$ is the number of correct predictions for class $c$ divided by the number of samples for class $c$. For a pruning ratio $t$, let $R_t^c(m)$ denote the recall for class $c$ of model $m$ on the test data after pruning.

We are particularly interested in the how these metrics differ for each class. Namely, let

$$B_t^c(m) = R_t^c(m) - A_t(m)$$

denote the *recall balance*[3]. When $B_t^c(m) > 0$, we say that model $m$ at pruning ratio $t$ *overperforms* for class $c$; and when $B_t^c(m) < 0$ we may say that $m$ *underperforms* for class $c$. Finally, let

$$\bar{B}_t^c(m) = \frac{B_t^c(m)}{A_t(m)} = \frac{R_t^c(m) - A_t(m)}{A_t(m)}$$

denote the *normalized recall balance*. The further away this value is from 0, the more pronounced is the difference in performance between class $c$ and the other classes in model $m$ at the pruning ratio $t$.

**Proposition 4.1.** *If a dataset with set of classes $\mathbb{C}$ is* balanced, *meaning that each class $c \in \mathbb{C}$ has the same number of samples, then for any model $m$ and pruning ratio $t$ it holds that*

$$\sum_{c \in \mathbb{C}} B_t^c(m) = 0.$$

*Proof.* If the dataset is balanced, then it follows that $A_t = \dfrac{\sum_{c \in \mathbb{C}} R_t^c}{|\mathbb{C}|}$ and hence $\sum_{c \in \mathbb{C}} R_t^c = |\mathbb{C}|A_t$. Therefore, $0 = \sum_{c \in \mathbb{C}}(R_t^c) - |\mathbb{C}|A_t = \sum_{c \in \mathbb{C}}(R_t^c - A_t) = \sum_{c \in \mathbb{C}} B_t^c$. $\qquad\square$

**Corollary 4.1.1.** *If a dataset is balanced, then for any model $m$ and pruning ratio $t$ it holds that*

$$\sum_{c \in \mathbb{C}} \bar{B}_t^c(m) = 0.$$

*Proof.* Immediate from dividing $\sum_{c \in \mathbb{C}} B_t^c(m)$ by $A_t(m)$ due to Proposition 4.1. $\qquad\square$

**Example**   Consider a model in which $A = 80\%$ with two classes $X$ and $Y$ such that $R^X = 90\%$ and $R^Y = 70\%$, and after pruning we have $A_t = 60\%$, $R_t^X = 70\%$, and $R_t^Y = 50\%$. Note that the recall balance remains the same for each class before and after pruning: $B^X = B_t^X = 10\%$ and $B^Y = B_c^Y = -10\%$. However, the predictions of the pruned network for class $Y$ are as good as a guess. In turn, the normalized recall balance is the same in absolute value when the two classes are compared, but that value increases due to pruning: $\bar{B}^X = -\bar{B}^Y = \frac{1}{8}$ and $\bar{B}_t^X = -\bar{B}_t^Y = \frac{1}{6}$.

Our choice of normalizing through dividing by $A_t(m)$ rather than by $R_t^c(m)$ aims at avoiding outliers from small recall values as well as undefined results if a high pruning ratio leads to zero recall. Moreover, an extension of Proposition 4.1 such as Corollary 4.1.1 would not be possible otherwise.

We may obtain similar properties by replacing recall $R^c(m)$ with *precision* $P^c(m)$. In other words, $P^c$ is the number of correct predictions for class $c$ divided by the total number of predictions for class $c$. By extension we may also consider the effect of pruning on the *F-score*, in which case we would consider replacing $R^c$ with the harmonic mean of recall and precision, i.e., $2\dfrac{R^c P^c}{R^c + P^c}$. However, we have preliminarily found the effect of pruning more expressive on recall than on precision or F-score.

---

[3]For a web search on April 22, 2022, there were no hits for either "recall minus accuracy" or "subtract accuracy from recall" and very few references for "difference between recall and accuracy".

# 5 Measuring intensification

For a given pruning ratio $t$ and class $c$ of a model $m$, we consider the following *intensification ratio*:

$$I_t^c(m) := \frac{\bar{B}_t^c(m)}{\bar{B}^c(m)} \equiv \frac{\text{Normalized recall balance \textbf{after} pruning}}{\text{Normalized recall balance \textbf{before} pruning}}$$

This metric can be used to evaluate if pruning widens the performance gap between classes, which happens when $I_t^c > 1$. For a class $c$ in which the original model overperforms ($\bar{B}^c > 0$), a ratio greater than 1 after pruning implies that $|\bar{B}_t^c| > |\bar{B}^c|$ since $\bar{B}^c > 0 \land I_t^c > 1 \rightarrow \bar{B}_t^c > \bar{B}^c > 0$. Similarly, for a class $c$ in which the original model underperforms ($\bar{B}^c < 0$), a ratio greater than 1 after pruning *also* implies that $|\bar{B}_t^c| > |\bar{B}^c|$ since $\bar{B}^c < 0 \land I_t^c > 1 \rightarrow \bar{B}_t^c < \bar{B}^c < 0$. In other words, we can use the same metric to determine if pruning comparatively improves and worsens normalized recall balance for classes in which the model respectively overperforms and underperforms.

**Example (cont.)**  For both classes $X$ and $Y$, we obtain the same intensification ratio $I_t^X = I_t^Y = \frac{4}{3}$. Hence, pruning the model leads to a greater disparity in relative performance for those classes.

Since $I_t^c = \frac{B_t^c}{B^c} \frac{A}{A_t}$ and we may assume $A_t < A$ for sufficiently large $t$, one could argue that $\frac{A}{A_t} > 1$ alone leads to $I_t^c > 1$. However, our experiments show quite the opposite in the case of random pruning, which can be regarded as a less careful selection of weights to prune. Moreover, we consider a ratio above 1 from another perspective. Namely, $I_t^c > 1$ implies $\frac{B_t^c}{B^c} > \frac{A_t}{A}$ indicating the relative drop in recall balance (if indeed it drops at all) is less severe than the relative drop in accuracy.

We summarize intensification across classes for a model $m$ at pruning ratio $t$ by estimating the slope $\alpha_t(m)$ of a simple linear regression between normalized recall balance before and after pruning:

$$\bar{B}_t^c(m) = \alpha_t(m)\bar{B}^c(m)$$

The slope $\alpha_t(m)$ is fit using all classes in $\mathcal{C}$. We omit the intercept term from this regression as it would always be zero in the balanced datasets that we study due to Corollary 4.1.1. Consequently, the ordinary least squares estimate of the slope becomes a weighted mean of the intensification ratios $I_t^c$:

$$\alpha_t = \frac{\sum\limits_{c \in \mathbb{C}} \bar{B}^c \bar{B}_t^c}{\sum\limits_{c \in \mathbb{C}} (\bar{B}^c)^2} = \frac{\sum\limits_{c \in \mathbb{C}} (\bar{B}^c)^2 I_t^c}{\sum\limits_{c \in \mathbb{C}} (\bar{B}^c)^2}$$

In this weighted mean, less weight is given to classes in which normalized recall balance before pruning is closer to 0. We believe this is an appropriate model-level summary, since it de-emphasizes classes for which recall and accuracy are nearly the same before pruning. Hence, we conclude that pruning induces an overall intensification on $m$ if $\alpha_t(m) > 1$ and de-intensification if $\alpha_t(m) < 1$.

In addition, within each model there is a statistical dependency between the set of per-class ratios $\{I_t^c \mid c \in \mathbb{C}\}$ obtained after pruning. However, the model-level summaries $\alpha_t(m)$ are independent across random initializations associated with training each model $m$ (conditional on the dataset), which allows us to use simpler statistical inference methods as described in Section 6.2. In our statistical analysis, we evaluate $\mathbb{E}[\alpha_t]$ as the hypothetical expectation across infinitely many $m$.

Comparatively, the statistical analysis in Hooker et al. [30] would be equivalent to evaluating $\mathbb{E}[I_t^c]$, as it is based on $\frac{R_t^c}{A_t} = I_t^c - 1$. A multiple linear regression with $\frac{R_t^c}{A_t}$ as dependent variable and $\frac{R^c}{A}$ as one of the independent variables is fit by Paganini [55] over a large number of models and classes.

# 6 Experimental setting

We seek to understand when $\mathbb{E}[\alpha_t] > 1$, or alternatively $\mathbb{E}[\alpha_t] < 1$, by evaluating the impact of network pruning algorithms on models trained on seemingly balanced datasets and through varying but well-known architectures. Hence, we consider the dependence of the intensification ratio at the model level by denoting it as $\alpha_{t,P}^{D,M}$ for a dataset $D \in \mathbb{D}$, an architecture $M \in \mathbb{M}$, a pruning ratio $t \in \mathbb{T}$, and a pruning algorithm $P \in \mathbb{P}$. In what follows, we may omit the indices if they are constant.

## 6.1 Computational details

We use some combinations of (i) models based on ResNet-$\{20, 32, 44, 56, 110\}$ [26] and an adaptation of LeNet5 [40] (ii) trained in MNIST [40], Fashion-MNIST [75], CIFAR-10, and CIFAR-100 [38], and then (iii) pruned using MP, GP, and Random Pruning (RP) from ShrinkBench [9] and our implementation of UP over ShrinkBench, all of which tested with (iv) pruning ratios 2, 4, 10, 20, and 50. For each combination of model and dataset evaluated, we have trained 30 models, which are the same used for pruning at each pruning ratio. The datasets were chosen due to their popularity and equal representation across classes. The architectures and pruning ratios were chosen based on preliminary experiments aiming for good accuracy and also to prune to up to a ratio with evidence of distortion. MNIST and Fashion-MNIST are only trained in LeNet5. When the dataset or model does not vary, we use CIFAR-10 and ResNet-56 due to their intermediary complexity. When the pruning algorithm does not vary, we use MP due to its popularity and seemingly better performance than GP [9].

The LeNet5 models are trained with SGD optimizer for 30 epochs, with batch size of 128 and learning rate of $0.01$, and then fine-tuned for another 15 epochs after pruning. The ResNet models are trained with SGD optimizer for 60 epochs, with batch size of 128, a decreasing learning rate schedule, and weight decay of $0.0005$. These values were selected based on preliminary testing. For all models, the weights for the epoch with the lowest validation loss are saved during training and fine-tuning for testing. All experiments are run on GPUs of Nividia GeForce 3060ti and 3090.

## 6.2 Statistical methods

We are interested in comparing the distributions of $\alpha_{t,P}^{D,M}$ across different scenarios. Each $\alpha_{t,P}^{D,M}$ is calculated using the test set recall metrics from just one run of training and a pair of models, corresponding to before and after pruning at ratio $t$. Thus we frame our statistical analyses as inferences about $\mathbb{E}[\alpha_{t,P}^{D,M}]$, where the expectation is taken over neural network models trained with different random seeds. We have tested varying $M$, $D$, or $P$ along with $t$. Hence, we evaluate the impact of the pruning ratio along every other dimension. For model complexity through $\alpha_t^M$, we use CIFAR-10 trained on ResNet-$\{20, 32, 44, 56, 110\}$ and pruned with MP. For dataset complexity through $\alpha_t^D$, we use all the datasets with their default model and pruned with MP. For pruning algorithm through $\alpha_{t,P}$, we use CIFAR-10 trained on ResNet-56 and pruned with all pruning methods. Due to the encouraging results with UP, we have repeated the model complexity with UP for comparison.

**Figures** Each scatterplot matrix shows the normalized recall balances before vs after pruning, at several $t$ (rows) and several of either $M$, $D$, or $P$ (columns). The scatterplots for varying $P$ are in Figure 5; the remaining scatterplots can be found in Appendix B, including Figure 8 for varying $M$ with UP instead of MP as the pruning algorithm. Each point corresponds to one class $c$ for one model $m$. One regression line and numeric summaries are overlaid in each subplot: $\hat{\alpha}$ is an average slope using all $m$ together, $r^2$ is the corresponding coefficient of determination, and $\bar{A}$ is the average accuracy across all $m$. Boxplots summarize how the corresponding model-level slopes $\alpha_{t,P}^{D,M}(m)$ vary across $m$, and how their distribution changes with $t$ and either $M$, $D$, or $P$. These boxplots are in Figures 1 to 3, in addition to Figure 4 for varying $M$ with UP instead of MP.

**Confidence intervals** At each boxplot, we calculate a t-based 99% confidence interval (CI) for its $\mathbb{E}[\alpha_{t,P}^{D,M}]$. We chose 99% confidence to achieve 95% family-wide confidence (within each family of 5 pruning ratios at a given $M$, $D$, or $P$) after a Bonferroni correction for simultaneously reporting 5 dependent CIs. The labels for pruning ratios in the boxplots include $<$, $>$, or ? to denote whether the corresponding CI is below 1 (de-intensification), above 1 (intensification), or overlaps 1. Appendix C has plots for each CI. Figures 1, 2, and 3 indicate that within each $M$, $D$, or $P$, low ratios tend to have CIs entirely below 1, some moderate ratios have CIs that overlap with 1, and high ratios tend to have CIs entirely above 1. The exception is $P$=RP, where the CIs are below 1 at all ratios.

**Hypothesis tests by $t$, $M$, $D$, or $P$** We carry out t-tests for each set of hypotheses in Section 7. Tests comparing pairs of $M$ or of $D$ are independent-samples tests, but tests comparing pairs of $t$ or of $P$ are paired-samples tests: for each $m$ there is a natural pairing between two $\alpha$s using the same uncompressed $m$. Each p-value reported in Tables 1 through 9 (Appendix A) has been multiplied

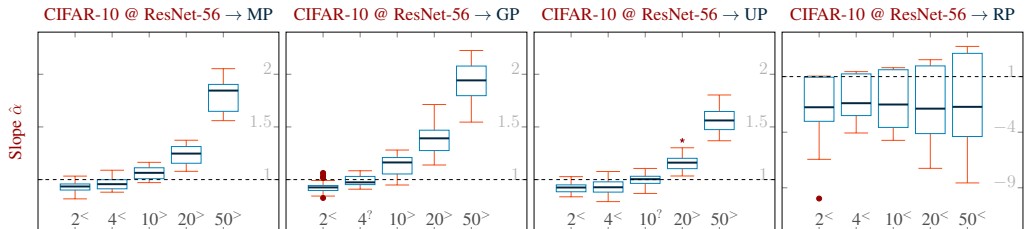

Figure 1: Boxplots of $\alpha_{t,P}^{D,M}(m)$ across $m$, at each $t$ within each $P$. Superscripts $<$, $>$, or ? denote where 99% CIs were below 1, above 1, or overlapped 1.

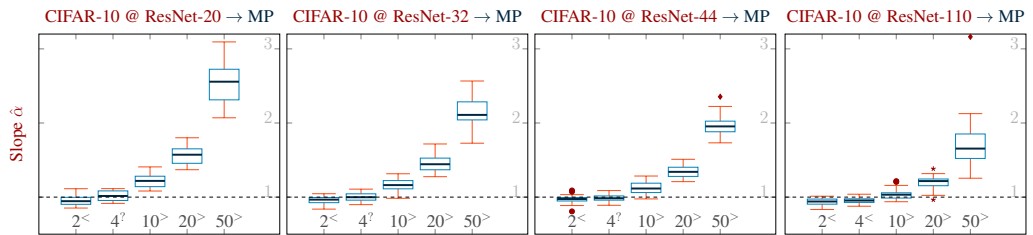

Figure 2: Boxplots of $\alpha_{t,P}^{D,M}(m)$ across $m$, at each $t$ within each $M$ for $P$ = MP. Superscripts $<$, $>$, or ? denote where 99% CIs were below 1, above 1, or overlapped 1.

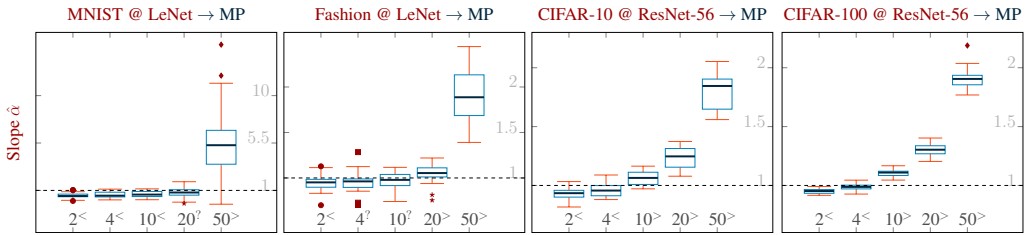

Figure 3: Boxplots of $\alpha_{t,P}^{D,M}(m)$ across $m$, at each $t$ within each $D$. Superscripts $<$, $>$, or ? denote where 99% CIs were below 1, above 1, or overlapped 1.

by the number of rows in its column of the table, as a Bonferroni multiple-testing correction for simultaneously evaluating all the rows in that column. When a reported p-value is below the usual significance level 0.05, we have evidence that intensification differs between the pairs in that setting.

**Accuracy before pruning** In order to assess the impact of pruning on accuracy, the mean accuracy before pruning for each dataset and model is reported in Table 10 (Appendix D).

## 7 Analysis

### 7.1 The influence of the pruning ratio

We carry out one-sided paired-samples t-tests between the following pairs of hypotheses:

$$H_0^{M,t,i} : \mathbb{E}[\alpha_{t_i}^M] \geq \mathbb{E}[\alpha_{t_{i+1}}^M] \qquad H_0^{D,t,i} : \mathbb{E}[\alpha_{t_i}^D] \geq \mathbb{E}[\alpha_{t_{i+1}}^D] \qquad H_0^{P,t,i} : \mathbb{E}[\alpha_{t_i}^P] \geq \mathbb{E}[\alpha_{t_{i+1}}^P]$$

$$H_a^{M,t,i} : \mathbb{E}[\alpha_{t_i}^M] < \mathbb{E}[\alpha_{t_{i+1}}^M] \qquad H_a^{D,t,i} : \mathbb{E}[\alpha_{t_i}^D] < \mathbb{E}[\alpha_{t_{i+1}}^D] \qquad H_a^{P,t,i} : \mathbb{E}[\alpha_{t_i}^P] < \mathbb{E}[\alpha_{t_{i+1}}^P]$$

In all of those, $t_i < t_{i+1}$ are consecutive pruning ratios in $\mathcal{T}$. We based our a priori on the observation that a higher pruning ratio typically leads to models that have a lower performance. As such we believe more pruned models would have less capacity to identify as many classes with similar recall when compared with less pruned models, and thus be more susceptible to intensification after pruning.

Figures 2 and 6 suggest that at each $M$, intensification tends to remain similar or become stronger at higher pruning ratios. Tables 1 and 2 confirm strong evidence of this trend between most consecutive

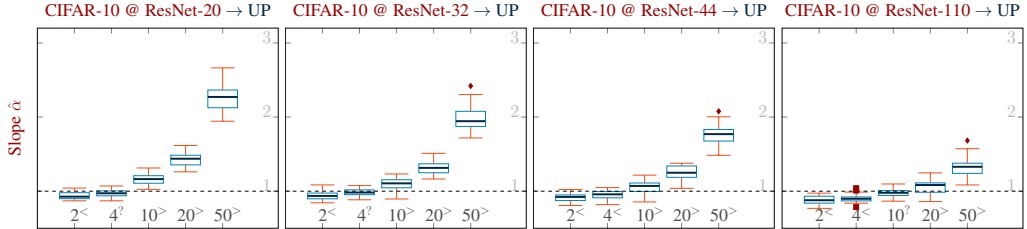

Figure 4: Boxplots of $\alpha_{t,P}^{D,M}(m)$ across $m$, at each $t$ within each $M$ for $P$=UP. Superscripts $<$, $>$, or $?$ denote where 99% CIs were below 1, above 1, or overlapped 1.

pairs of ratios, for both MP and UP, at most model sizes, except for ratios 2 vs 4 on some larger architectures.

Figure 3 and Figure 7 suggest that at each $D$, intensification tends to remain similar or become stronger at higher pruning ratios. Table 3 confirms that we have strong evidence of increasing intensification for MNIST when going from pruning ratio 20 to 50; for Fashion, when going from 10 to 20 and from 20 to 50; and for both CIFAR-10 and CIFAR-100, for every consecutive pair of ratios.

Figure 1 and Figure 5 suggest that at each $P$ except RP, intensification tends to remain similar or become stronger at higher pruning ratios. Table 4 confirms strong evidence of this trend between most consecutive pairs of ratios at most $P$. However, for RP, the average slope actually appears to decrease instead during several ratio increases.

## 7.2 The influence of model complexity

For each ratio $t \in \mathcal{T}$, we carry out a one-sided independent-samples t-test of

$$H_0^{M,i,t} : \mathbb{E}[\alpha_t^{M_i}] \leq \mathbb{E}[\alpha_t^{M_{i+1}}]$$

$$H_a^{M,i,t} : \mathbb{E}[\alpha_t^{M_i}] > \mathbb{E}[\alpha_t^{M_{i+1}}]$$

for $M_i$ and $M_{i+1}$ as consecutive models in ResNet-$\{20, 32, 44, 56, 110\}$ on CIFAR-10, using MP and UP. We based our a priori on the observation that larger (deeper) networks are able to learn more complex nonlinear functions than smaller (shallower) networks. As such we believe smaller pruned models would have less capacity to identify as many classes with similar recall when compared to larger pruned models, and thus be more susceptible to intensification after pruning.

Figures 2, 4, 6, and 8 suggest that at most ratios, smaller model sizes tend to have more intensification than larger model sizes. Tables 5 and 6 confirm strong evidence of this trend between most consecutive pairs of model sizes (except 56 vs 110 with MP) at high $t$, but rarely at low $t$.

## 7.3 The influence of dataset complexity

For each ratio $t \in \mathcal{T}$, we carry out a one-sided independent-samples t-test of

$$H_0^{D,i,j,t} : \mathbb{E}[\alpha_t^{D_i}] \geq \mathbb{E}[\alpha_t^{D_j}]$$

$$H_a^{D,i,j,t} : \mathbb{E}[\alpha_t^{D_i}] < \mathbb{E}[\alpha_t^{D_j}]$$

for the following $(D_i, D_j)$ pairs: (MNIST, CIFAR-10); (Fashion, CIFAR-10); and (CIFAR-10, CIFAR-100), using MP. We based our a priori on how "more complex" datasets often require larger (deeper) models to achieve acceptable accuracy. As such we believe by increasing dataset complexity the pruned model would not be able to identify as many classes with similar recall when compared to less complex datasets, and thus be more susceptible to intensification. We judged that MNIST and Fashion have comparable complexity (10 black-and-white classes), but both are less complex than CIFAR-10 (10 RGB classes), which is less complex than CIFAR-100 (100 RGB classes).

Figure 3 and Figure 7 suggest that at most ratios, MNIST and Fashion tend to have shallower slopes (less intensification) than CIFAR-10, which has shallower slopes than CIFAR-100. Table 7 confirms strong evidence of this trend between MNIST and CIFAR-10 at the smaller ratios; between Fashion

and CIFAR-10 only at moderate ratios; and between CIFAR-10 and CIFAR-100 only at the higher ratios.

## 7.4 The influence of the pruning algorithm

For each ratio $t \in \mathcal{T}$, we carry out a two-sided paired-samples t-test of

$$H_0: \qquad \mathbb{E}[\alpha_t^{P_i}] = E[\alpha_t^{P_j}]$$

$$H_a: \qquad \mathbb{E}[\alpha_t^{P_i}] \neq E[\alpha_t^{P_j}]$$

for pruning algorithms $P_i, P_j \in \{MP, GP, UP, RP\}$ on CIFAR-10 with ResNet-56. Without clear a priori expectations for which algorithms would experience more intensification than others, we compare all pairs and use two-sided tests. We also compared just MP and UP at all models in ResNet-$\{20, 32, 44, 56, 110\}$.

Figure 1 and Figure 5 suggest a trend in which, at most ratios, UP tends to have shallower but still positive slopes (less intensification) than MP or GP, while RP always has negative slopes. Table 8 confirms we have evidence that the average slopes for MP, GP, and UP differ from each other at higher pruning ratios, but not necessarily at the lowest ratios. We also have strong evidence that all three methods differ from RP, at each ratio. Moreover, comparing Figure 8 to Figure 6 suggests that UP tends to have less intensification than MP at each $t$ and $M$. Table 9 confirms this for the larger ratios and models.

We can also observe by comparing Figure 5 and Figure 8 that $\hat{\alpha}$ is always smaller for UP, whereas $\bar{A}$ is only slightly greater for MP in 3 out of 20 cases: ResNet-20 with $t = 4$, ResNet-110 with $t = 2$, and ResNet-110 with $t = 4$.

# 8 Supplemental discussion and experiments

Following the advice from the anonymous reviewers, we have augmented our discussion and the experiments carried out in the paper. First, we consider the tradeoff between accuracy and intensification at lower pruning ratios to discuss the operationalization of our results in Appendix E. Second, we report the variance of recall balance in Appendix F. Third, we discuss at greater length the use of $\alpha$ as our metric of interest in Appendix G. Finally, we included additional results comparing accuracy and intensification for two recent approaches, LTH [17] and CHIP [65], in Appendix H.

# 9 Conclusion

In this work, we have found evidence that network pruning may cause recall distortion even in models trained on seemingly balanced datasets, thereby complementing prior studies with underrepresented classes and features [30, 55, 31, 1, 63, 69]. In our experiments, we have observed a statistically significant effect of higher pruning ratios, increased task difficulty, and greater model complexity intensifying the effect of normalized recall balance — the difference between recall and accuracy divided by accuracy. This happens for both underperforming classes (recall below accuracy) and overperforming classes (recall above accuracy) for any pruning method other than Random Pruning (RP), hence showing that accuracy reduction alone does not lead to intensification. More surprisingly, however, is that we have observed the opposite phenomenon at lower pruning ratios. Namely, we have observed a de-intensification effect, by which model accuracy improves at the same time that the differences between class recall and model accuracy goes down. To the best of our knowledge, no other work has identified this corrective effect as a positive externality of moderate pruning.

We have also introduced Undecayed Pruning (UP) as a variant of Gradient Pruning (GP) that discounts the effect of weight decay from the loss function, leading to an algorithm that is more closely aligned with the better acclaimed Magnitude Pruning (MP). Since UP consists of a fine adjustment between the MP and GP approaches, we are not surprised that this leads only to a minor improvement. Nevertheless, it shows that a better alignment of the pruning method with the loss function alleviates recall balance. We may explore in future how to deduct other forms of regularization before pruning.

One possible limitation of our study is that it does not include a vast list of pruning algorithms, which would nevertheless be prohibitive for a comprehensive statistical evaluation. However, it emphasizes

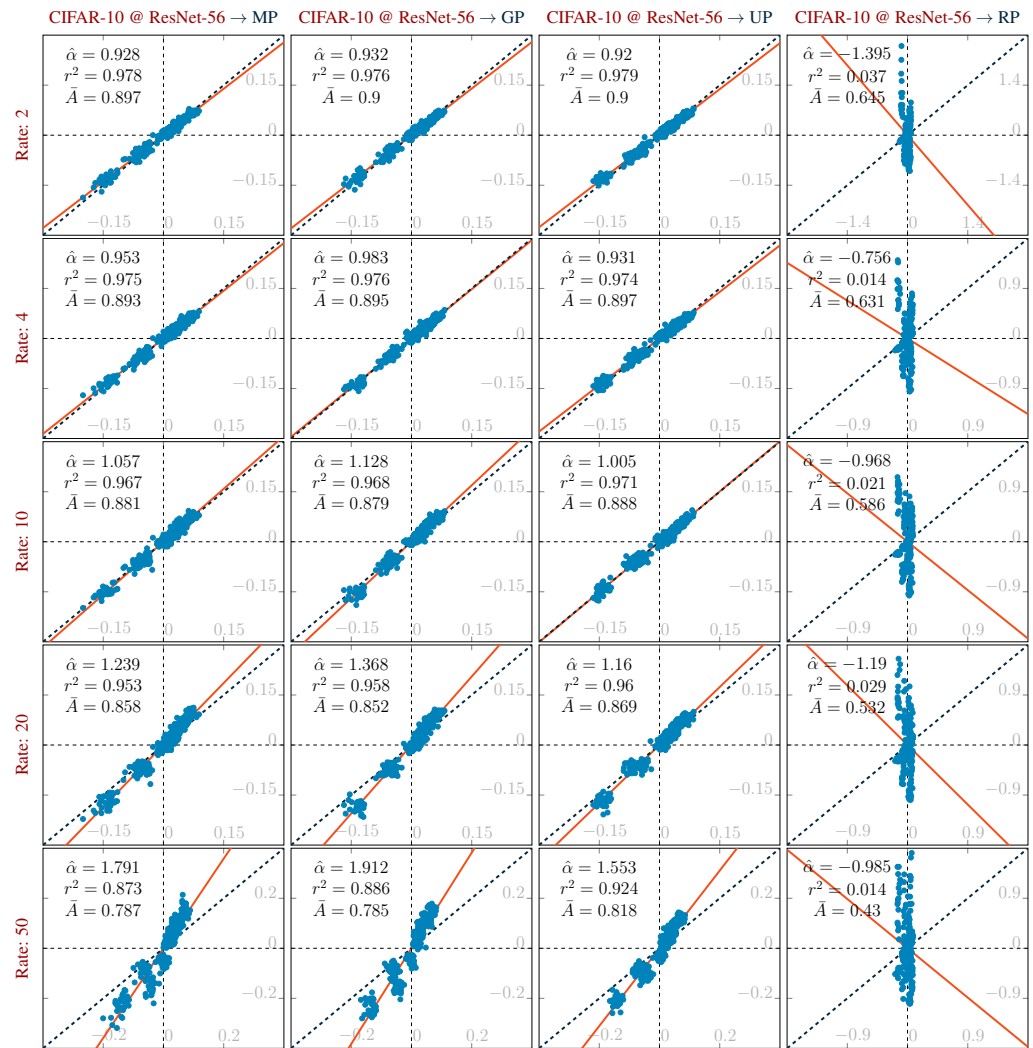

Figure 5: Scatterplot matrix of $\bar{B}^c(m)$ ($x$-axis) vs $\bar{B}_t^c(m)$ ($y$-axis), at several values of $t$ (rows) and $P$ (columns). Each scatterplot point corresponds to one $c$ for one $m$. See Section 6.2.

the classic methods conventionally used and sheds light on how they can be combined through our new algorithm. Nevertheless, we believe that the insight on how to develop better variations of simple algorithms such as MP and GP helps us understand how to design more complex network pruning algorithms as well. In fact, additional experiments carried out by suggestion of the anonymous reviewers show that the intensification effect can also be observed in modern pruning methods.

## Acknowledgments and Disclosure of Funding

We would like to thank the anonymous reviewers for the constructive discussion, including their suggestions to expand the scope of our work as well as to consider future directions of work.

Aidan Good, Jiaqi Lin, Hannah Sieg, and Thiago Serra were supported by the National Science Foundation (NSF) grant IIS 2104583. Hannah Sieg was supported by the H. Royer Undergraduate Research Fund. Mikey Ferguson was supported by Bucknell University's Presidential Fellowship. Xin Yu was partially supported by the National Science Foundation (NSF) grant IIS 1764071. Shandian Zhe was supported by the National Science Foundation (NSF) grant IIS 1910983.

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
