# OpenReview forum: "Recall Distortion in Neural Network Pruning and the Undecayed Pruning Algorithm"
_NeurIPS.cc/2022/Conference — NeurIPS 2022 Accept_

### Official Review · Reviewer_imMk · 2022-07-13

**Rating:** 5
**Confidence:** 2
**Soundness:** 2 fair
**Presentation:** 3 good
**Contribution:** 2 fair

**Summary:**

This paper claims that network pruning inherently makes the model's class-wise accuracy (i.e., recall) imbalanced across classes, which the authors call intensification effect. In order to prove the hypothesis, some ratios are introduced, which models relative performance of each class with respect to the overall accuracy, and thereby intensification ratio is finally defined and experimentally measured. In the experiments, intensification effect is examined in terms of model complexity and dataset complexity, and the corresponding results show that the more complex the dataset, the simpler the model, the stronger the intensification effect.

**Questions:**

1. Please refer to Weakness 3 above.
2. How is the variance of class-wise recalls? Does it become larger with a higher pruning ratio?


**Limitations:**

Yes.

**Strengths And Weaknesses:**

(Strengths)
1. The paper is well written and motivated, providing novel insights on network pruning.
2. The hypotheses are adequately designed and analyzed in a statistical manner.
3. Undecayed pruning is devised via a novel finding about the relationship between magnitude pruning and gradient pruning.

(Weaknesses)
1. As the authors also mentioned, the tackled pruning schemes are somewhat naive and conventional, compared to the SOTA pruning methods. This weakness makes this work less interesting in practice.
2. Some new metrics should be more rigorously justified. For example, recall balance and intensification ratio can be defined in many other ways rather than computing the relative figure with respect to the overall accuracy.
3. In Sections 7.2 and 7.3, the expected results on intensification effect are not well explained. Thus, why the smaller models are prone to intensification effect? and why the more complex datasets can make the effect stronger?

---

> ### Author Response · Authors · 2022-08-02
> **Effect of pruning ratio on recall variance**
>
> The table below shows the average recall variance before and after pruning for each type of model and pruning ratio used in our study. All the cases in which the recall variance after pruning is greater than before pruning are in bold. That probably confirms the reviewer’s intuition that recall variance increases with pruning. We would emphasize, however, that variance alone is not a clear indicator if recall differences increase or decrease.
>
> | Model | Before | Ratio 2 | Ratio 4 | Ratio 10 | Ratio 20 | Ratio 50 |
> | :---: | :---: | :---: | :---: | :---: | :---: | :---: |
> | CIFAR-10 @ ResNet-20 → MP | 0.0036 | **0.0036** | **0.0039** | **0.0057** | **0.0097** | **0.0318** |
> | CIFAR-10 @ ResNet-32 → MP | 0.0032 | 0.0031 | **0.0033** | **0.0045** | **0.0074** | **0.0189** |
> | CIFAR-10 @ ResNet-44 → MP | 0.0038 | 0.0035 | 0.0037 | **0.0048** | **0.0068** | **0.0153** |
> | CIFAR-10 @ ResNet-110 → MP | 0.0039 | 0.0036 | 0.0037 | **0.0045** | **0.0061** | **0.0135** |
> | CIFAR-100 @ ResNet-56 → MP | 0.0004 | 0.0004 | 0.0004 | **0.0005** | **0.0009** | **0.0021** |
> | Fashion @ LeNet → MP | 0.0099 | 0.0098 | 0.0088 | 0.0088 | **0.0115** | **0.0416** |
> | MNIST @ LeNet → MP | 0.0038 | **0.0038** | **0.0038** | **0.0038** | **0.0039** | **0.0111** |
> | CIFAR-10 @ ResNet-56 → MP | 0.0038 | 0.0034 | 0.0036 | **0.0045** | **0.0062** | **0.0142** |
> | CIFAR-10 @ ResNet-56 → GP | 0.0039 | 0.0033 | 0.0037 | **0.0048** | **0.0072** | **0.0149** |
> | CIFAR-10 @ ResNet-56 → UP | 0.004 | 0.0034 | 0.0033 | **0.004** | **0.0056** | **0.0104** |
> | CIFAR-10 @ ResNet-56 → RP | 0.0038 | **0.1504** | **0.132** | **0.1413** | **0.1487** | **0.227** |

---

> ### Author Response · Authors · 2022-08-02
> **Justifying the metrics**
>
> This is an interesting point, and we break it down into each step below.
>
> **Recall**: To the best of our knowledge, we have not seen the term recall explicitly discussed previously in this literature. However, at least four papers that are directly relevant to us  deal with exactly the same concept under different terminologies, such as class level accuracy [27,52], error per class [28], and class sample rate [7]. In all of these cases, their concern is similar to ours: that the recall for some classes is more severely affect than for others, in which case the model has disparate performance across classes.
>
> **Recall balance**: From the discussion above, we believe that it is only natural to measure the difference between recall and accuracy if we want to distinguish between classes for which the model performs better or worse than average, which is the same concern as in prior work.
>
> **Normalization**: We do take a leap, which we believe is justified, in calculating the ratios of normalized recall balances. As we discuss in the paper, recall balance alone is not a good metric for comparison if the accuracy is also affected by network pruning. After all, it is more intuitive to care about the difference in proportion to the accuracy, as illustrated by the example in pages 4 and 5 of our manuscript.
>
> **Intensification**: From that point, the intensification as a ratio of normalized recall balance after and before pruning is a clear form of quantifying if pruning makes such differences greater or smaller than before.

---

> ### Author Response · Authors · 2022-08-02
> **Expected results of the experiments**
>
> In Section 7.2, the purpose of the test is to evaluate if the intensification is greater in smaller models. We based our a priori on the observation that larger (deeper) networks are able to learn more complex nonlinear functions than smaller (shallower) networks. As such we believe smaller models would have less capacity to identify as many classes with similar recall when compared to larger models. Consequently, smaller models would be more susceptible to intensification. While our experiment was not to determine if this mechanism is actually occurring, this was our reasoning for the hypothesis direction chosen.
>
> In Section 7.3, the purpose of the test is to evaluate if the intensification is greater when a dataset is more complex, such as having larger dimensions or more classes. We based our a priori on how “more complex” datasets often require larger (deeper) models to achieve acceptable accuracy. As such we believe by increasing dataset complexity the model would not be able to learn to identify as many classes with similar recall when compared to less complex datasets. Consequently, models trained on more complex datasets would be more susceptible to intensification. While our experiment was not to determine if this mechanism is actually occurring, this was our reasoning for the hypothesis direction chosen.

---

> ### Author Response · Authors · 2022-08-02
> **Comparison with SOTA pruning algorithms**
>
> Following your recommendation and some suggestions from reviewer hKwV, we have extended our analysis to a few recent papers.
>
> We do note, however, that most of the current approaches to network pruning either derive from MP or from GP. For example, the original approach in the Lottery Ticket Hypothesis (LTH) paper is based on MP [16]. Many approaches based on the functional expansion of the loss function can similarly be understood as exensions of GP. Therefore, we believe that centering our analysis on such fundamental approaches would be preferable to focusing on recent variations of those principles. Nevertheless, we see that added value of including SOTA algorithms as a means to evaluate if recall distortion has also been addressed by them.
>
> Based on our preliminary findings, it does seem that new approaches have been successful in reducing intensification, although the results below are based on a single model on which different compression ratios are applied. We will update these results with more models this week.
>
> The following results are obtained using LTH with 10k and 64k steps of training at each step of the sparsification process. We note that with fewer steps we do not observe intensification, whereas the model accuracy reaches better values with more steps.
>
> | Step | LTH 10k Accuracy | LTH 10k Intensification | LTH 64k Accuracy | LTH 64 Intensification |
> | :---: | :---: | :---: | :---: | :---: |
> | 0 | 0.81 | - | 0.87 | - |
> | 1 | 0.83 | 0.71 | 0.88 | 0.86 |
> | 2 | 0.83 | 0.63 | 0.89 | 0.75 |
> | 3 | 0.83 | 0.80 | 0.89 | 0.73 |
> | 4 | 0.84 | 0.73 | 0.86 | 1.08 |
> | 5 | 0.84 | 0.65 | 0.87 | 0.72 |
> | 6 | 0.84 | 0.56 | 0.86 | 0.84 |
> | 7 | 0.84 | 0.78 | 0.86 | 0.84 |
> | 8 | 0.84 | 0.64 | 0.86 | 0.79 |
> | 9 | 0.84 | 0.50 | 0.85 | 0.96 |
> | 10 | 0.83 | 0.82 | 0.83 | 1.10 |
> | 11 | 0.83 | 0.68 | 0.81 | 1.20 |
> | 12 | 0.83 | 0.67 | 0.83 | 0.91 |
> | 13 | 0.81 | 0.63 | 0.80 | 1.26 |
> | 14 | 0.80 | 0.76 | 0.78 | 1.41 |
> | 15 | 0.78 | 0.82 | 0.79 | 1.22 |
>
> We also refer the reviewer to our comparison with CHIP, as requested by reviewer hKwV (see post "Comparison with other pruning methods").

---

### Official Review · Reviewer_hKwV · 2022-07-13

**Rating:** 6
**Confidence:** 3
**Soundness:** 3 good
**Presentation:** 4 excellent
**Contribution:** 3 good

**Summary:**

Paper studies the problem of recall distortion in neural network pruning. Particularly, authors observe that pruning makes recall relatively worse for a class with recall below accuracy  and makes recall relatively better for a class with recall above accuracy. Authors proposed a new pruning algorithm namely undecayed pruning which can attenuate such effect. Observations made in the paper will be helpful to the future work on network pruning.

**Questions:**

Please see 1-3 in Weaknesses.

**Limitations:**

Yes

**Strengths And Weaknesses:**

**Strengths:**
1. Paper studies an important problem of recall distortion in neural network pruning.
2. Selected number of criteria is quite representative and contains magnitude based criteria, gradient based criteria and random pruning.
3. Proposed properties of the pruning are quite interesting and novel. For example, intensification is less severe with the proposed undecayed pruning algorithm but nevertheless more pronounced with relatively more difficult tasks, less complex models, and higher pruning ratios
4. Many pruning papers consider only one setting, but authors use the same framework (datasets, models, pruning ratios) and provide insights by rigorous statistical tests.

**Weaknesses:**
1. Paper lacks some conclusions about how to interpret intensification. In other words, how intensification helps design better pruning algorithms.
2. The proposed undecayed pruning is affected by the regularization techniques applied during training. For example, weight decay is often used in normal training. Batch Normalization and Dropout have implicit regularization. Could authors comment on how those settings will affect the undecayed pruning?
3. Other pruning methods: Significant progresses have been made recently in developing advanced network pruning techniques, such as [1, 2]. Do these methods have the same effect of recall distortion? Or are these methods already a stepping stone in the right direction towards reducing intensification?
[1] Minsoo Kang and Bohyung Han. Operation-aware soft channel pruning using differentiable masks. In ICML, 2020.
[2] Yang Sui et al. CHIP: CHannel independence-based pruning for compact neural networks. In NeurIPS 2021.

**Post rebuttal**

Thank you for your detailed rebuttal. All my concerns are addressed, it includes a comparison with other pruning methods. I decide to increase my rating from 5 to 6.

---

> ### Author Response · Authors · 2022-08-02
> **How knowing about intensification helps**
>
> As we have discussed with reviewer X7NS (see post “Reduction in recall distortion for small amounts of pruning & operationalization”), we are contributing with a new metric to evaluate network pruning algorithms. This metric can be used in addition to model accuracy to more thoroughly evaluate the side effect of a network pruning algorithm by avoiding a disproportionate effect in class recalls. Moreover, the results of our experiments indicate how to compensate for changes in comparative task difficulty through changes in model complexity and pruning ratios used. We replicate below some parts of our discussion that we believe are particularly useful.
>
> *[...] More specifically, we observe model accuracy improving at the same time that intensification is reduced if a small pruning ratios is used.*
>
> *We have attached some preliminary plots to the supplementary material (see file AIxR.png), in which we refine our study a lower pruning ratios by evaluating 30 models at ratios 2, 4, 6, 8, and 10. In those plots, the green curve associated with the left y-axis represents the mean accuracy at each compression ratio, with the initial observation at compression ratio 1 corresponding to the model accuracy before pruning. The blue curve associated with the right y-axis represents the mean intensification at each compression ratio. We have aligned both y-axes so that the center of the left axis represents the mean accuracy before pruning and the right axis represents an intensification ratio of 1, and then a black horizontal line is drawn at the center of the plot. Whenever we see the green plot above that line and the blue plot below that line, which is typical for the lower pruning ratios, we are observing accuracy going up while intensification is going down.*
>
> *[...] When the pruning ratio increases, model accuracy and intensification no longer move in the same direction. Whereas heavy pruning makes the accuracy worse and intensification stronger, lighter pruning makes the accuracy better while not pushing intensification above 1.*
>
> *We believe that these results are actionable to the extent that they encourage the use of a moderate amount of pruning both for the sake of improving generalization as well as to reduce the performance balance across classes. Moreover, our results provide a qualitative understanding of how to adjust for different cases. Namely, if we work with a comparatively more complex task we should compensate with a larger model or a lower pruning ratio if we would like to obtain a pruned model with similar improvements for both metrics.*

---

> ### Author Response · Authors · 2022-08-02
> **Accounting for other forms of regularization when pruning**
>
> We love this idea and we will look further into it. We replicate below a discussion with reviewer X7NS (see post “Motivation and benefit of UP”) to justify proposing UP for the case of networks trained with weight decay in the scope of this study.
>
> *The motivation for UP is to address what we believe is a double-edged approach to reducing the number of parameters. Namely, that it is common to use a term in the loss function to make the weights as small as possible, such as weight decay, and then that we use the gradient of the loss function for pruning weights while ignoring that the gradient is also affected by the weight decay term.*
>
> *It is currently accepted that magnitude-based pruning (MP) works better in practice than gradient-based pruning (GP), even though MP is based on the somewhat misleading assumption that smaller weights have a smaller impact if pruned (counterexamples exist since the 1980s [23,38]), whereas GP is more principled for being based on the effect of pruning the weight in the loss function.*
>
> *However, if we revisit GP with the premise that the weight decay term should be deducted from the loss function first, then we not only improve the results with GP but we also obtain better results than with MP. Moreover, the way in which the weights are ultimately ranked reflects a weighted combination of the criteria used for MP and GP. This is particularly interesting because UP coincides with MP when the gradients used for GP are sufficiently close to zero, hence providing a principled argument for the effectiveness of MP under weight decay.*
>
> *Since UP consists of a fine adjustment between the MP and GP approaches, we are not surprised that this leads only to a minor improvement. However, this improvement shows that a better alignment of the pruning method with the loss function alleviates recall balance. We believe that presenting this adjustment as part of the paper illustrates how recall balance issues can be addressed.*

---

> ### Author Response · Authors · 2022-08-02
> **Comparison with other pruning methods**
>
> We truly appreciate this suggestion, including the direct reference to methods that we have missed in our literature review. We were able to evaluate CHIP this week, but not your first reference due to some compatibility issues because that paper’s code uses older versions of libraries such as Tensorflow.
>
> The table below compares CHIP and MP for the compression ratios reported in the CHIP paper by using one model. We will expand this analysis to 30 models this week to make sure that the results are consistent.
>
> | Ratio | MP Accuracy | MP Intensification | CHIP Accuracy | CHIP Intensification |
> | :---: | :---: | :---: | :---: | :---: |
> | 0 | 0.89 | -- | 0.89 | -- |
> | 1.75 | 0.90 | 0.90 | 0.93 | 0.57 |
> | 3.33 | 0.90 | 0.94 | 0.92 | 0.57 |
>
> We also refer the reviewer to our comparison with LTH, based on the suggestion of reviewer imMk of evaluating SOTO methods (see post "Comparison with SOTA pruning algorithms").

---

### Official Review · Reviewer_X7NS · 2022-07-19

**Rating:** 7
**Confidence:** 4
**Soundness:** 4 excellent
**Presentation:** 3 good
**Contribution:** 3 good

**Summary:**

The authors study how pruning network parameters affects relative recall across classes. In particular, they argue that high levels of pruning increases the asymmetry in recall already present in the dense network. If recall for a class is below accuracy before pruning, it tends to decrease relative to accuracy after pruning, and if it is larger than accuracy before pruning, it tends to improve relative to accuracy after pruning.

They evaluate this hypothesis by constructing a model-level metric for “intensification” that aggregates information across classes and conducting statistical tests to verify this intensification effect. Using similar statistical tests, they observe that intensification is higher at high pruning ratios, and for high model and data complexity, but lower for low pruning ratios. They also study how the intensification effect varies across 3 existing pruning methods and introduce a new pruning method which they claim decreases this intensification effect.

**EDIT:** The authors have sufficiently addressed my main concerns and the additional experiments are quite promising and make this a stronger paper. The methodology in this paper is quite thorough and I think that the findings will be interesting to the community. I am raising my score from a 6 to 7.

**Questions:**

**First, minor points on clarity:**

The contributions section is not well worded and rather unclear. The paper would benefit from more proofreading of that section and a clearer, more explicit statement of the contributions. In particular:
1. Contribution (i) has grammatical errors that make it a little unclear. I also am not sure what exactly the contribution is here: the method of statistical analysis is not novel.
2. Contribution (ii) should be more explicit: what exactly is meant by "less severely affected" and "more severely affected"?

**Questions about $\alpha$:**

I think that there are some issues about $\alpha$ that need to be addressed since this is the basis for most of the experiments. I am not sure $\alpha$ is the best way to aggregate information across classes. As mentioned in the paper, it down-weights classes with small $\|R^c - A\|$ _before_ pruning. This seems problematic as it would down-weight the examples where there is small negative recall balance before pruning but large negative recall balance after pruning, which is a problematic case. I think that it is also important to include an ablation study of just using the simplest obvious metric: $I^c_t$ averaged across classes without weightings to make sure that the the results still hold.

**Significance:**

I would encourage the authors to be more explicit about what additional information this paper provides compared to previous work and how that is useful to the community.

Right now, the benefit of the newly introduced pruning method is unclear. The method does not seem very well motivated. Why is doing GP with this additional L2 term a useful thing to do? Why does it reduce intensification of the recall distortion? Also though UP does seem to do better than MP, the gains seem to be marginal. A more explicit and clearly presented head to head comparison and statistical significance of the gains would be helpful here.

Another route to impact would be to further explore the reduction of recall balance from small amounts of pruning. How significant is this finding and how sensitive is it to the fraction pruned? How does the overall accuracy change? Would the authors recommend this as a good practice to improve all models post training, and if so, why?

Another direction to explore would be: how can this finding of the connection between recall balance pre and post pruning be operationalized? Are there ways we can mitigate this effect? Can this be used for model selection between models that perform with similar accuracy, i.e. if two models have similar accuracy before pruning but one has less recall distortion, will both models have similar accuracy after pruning? If so, this could be used as a metric for selecting models with lower post pruning recall distortion. Is there a tradeoff between lower recall distortion and overall accuracy after pruning? Or perhaps, this new metric should be another axes along which pruning methods should be evaluated. In that case, a well documented and easy to use code base that would enable people to very easily and quickly adopt this metric into their analysis would be helpful.

I think that improving this work in 1 or more of these directions to really operationalize this would take it beyond an interesting observation and turn it into a strong and impactful work for the community.

**Limitations:**

The authors have sufficiently addressed the limitations of the work.

**Strengths And Weaknesses:**

**Strengths:**

Though previous work has demonstrated that the impact of pruning is uneven, as far as I know, the specific hypothesis that pruning intensifies a pre-existing recall distortion is novel. This is an interesting hypothesis and has potential impact as it enables one to predict, pre-pruning i.e. based on the dense model, which classes are likely to be more affected by pruning. The authors perform a thorough and systematic statistical analysis to verify their hypotheses. The authors also study under what conditions (variations in model/dataset complexity and pruning ratio) the hypothesis holds. The methodology is sound and commendable. Additionally, the finding that recall distortion at low pruning ratios decreases intensification is potentially very interesting and impactful. The paper is overall well written and clear, though a few minor typos and grammatical errors are a little distracting—I would recommend a careful proofread.


**Weaknesses:**

Despite the thorough experimentation, this work has a few weaknesses that I think needs to be further addressed. I will outline them here but please see the Questions section for a more thorough discussion.
1. The primary metric $\alpha$ used in this paper has some weaknesses.
2. The direct utility of this work is a little unclear. A more explicit discussion about how this work is different from previous work and what the authors think the marginal benefit of this work is would also be helpful.
   (a) The motivation for and benefit of UP is unclear.
   (b) Reduction in recall distortion from small amounts of pruning is under-explored.
   (c) Lacks a demonstration of how this can be operationalized.

---

> ### Author Response · Authors · 2022-08-02
> **The metric $\alpha$**
>
> We understand the reviewer’s concern with $\alpha$ not attributing the same weight to large variations around the origin. However, we note that across all scatterplots of ($x=\bar{B}^c$, $y=\bar{B}^c_t$) in the experimental data that we collected, we did not observe any behavior around the origin that would differ substantially from the linear trend. Hence, the slope (as measured by $\alpha$) seems to be an appropriate summary of trends seen in our plots. On the other hand, points near the origin correspond to intensification ratios whose denominators are near zero, and hence on the scale of y/x ratios they are often volatile outliers—even though on our scatterplots their (x,y) pairs are not outliers. Since the equally-weighted mean of intensification ratios is not robust to outliers, it is not an appropriate summary of trends seen in our data. In the revised paper, we will point this out clearly when we justify our use of alpha.
>
> To illustrate that, consider the results for CIFAR-10 @ ResNet-32 with MP at rate 20 (page 9, figure 3, row 4, column 2). We picked this plot because it has many points concentrated around the origin while presenting a clearly linear behavior. The value of $\alpha$ reported in the plot (1.455) corresponds to a linear regression using data from all models and classes. If we calculate $\alpha$ for each of the 30 models separately, we obtain 1.5 $\pm$ 0.1 with a minimum of 1.3 and a maximum of 1.7. In turn, if we calculate the mean of the intensifications averaging all classes for each model, we obtain 1.4 $\pm$ 1.4 with a minimum of -5.0 and a maximum of 4.5.
>
> For the model that yields the minimum of -5.0, there is a single outlier intensification ratio, corresponding to the pair of (x,y) values (0.0004, -0.0285). For the class associated with these values, the recall before pruning is 90.3% and the model accuracy is 90.26%, hence implying that the model overperforms for this class. Since the test set has 1,000 samples for each class, it would take only one more sample being incorrect for the model to underperform for this class. However, this class alone contributes with an intensification of -64, which would have been positive but similarly large in absolute value if one more test sample were incorrect before pruning.
>
> For the model that yields the maximum of 4.5, there is a single outlier intensification ratio, corresponding to the pair of (x,y) values (-0.0003, -0.0099). The recall for this class before pruning is again 90.3% and the model accuracy is 90.33%, hence implying that the model slightly underperforms for this class. Although the intensification in this case is 30, we note that the normalized recall balance after pruning remains the smallest across all classes. Furthermore, it would take only one more sample being incorrect for the model to overperform for this class, in which case the intensification would be negative but again similarly large in absolute value.
>
> In other words, we believe that the value of $\alpha$ represents a more consistent and representative characterization of the intensification effect of pruning on recall balance, since it reflects the consistent trends across models shown in our scatterplots. It is true that $\alpha$ gives less weight to outlier cases corresponding to classes that had recall very close to the model accuracy before pruning, but we believe this is reasonable because those classes have unstable ratios due to their denominators being near zero. Moreover, since their recalls are so close to the accuracy, it does not seem as appropriate to attribute such changes to intensification.
>
> Another way to think about this is that for the classes in such a situation, the model is only narrowly over- or underperforming, which means that the intensification ratio for the class is not as informative for our purposes. We agree that $\alpha$ is not the only way to aggregate information across classes, but we wish to emphasize that $\alpha$ and intensification ratios are meant to help us think about questions such as “If a recall-balance is already non-negligible, when does pruning push it even farther away in the same direction?” This is distinct from asking general questions about variability, such as “When does pruning make small recall-balances more variable?”

---

> ### Author Response · Authors · 2022-08-02
> **Minor points on clarity**
>
> Regarding contribution (i), we agree that the method of analysis is not novel. Hence, we should have used a verb such as “conduct” instead of “develop”, since we believe that performing such an extensive evaluation is a contribution in itself.
>
> Regarding contribution (ii), by “more severely affected” and “less severely affected” we mean the cases in which the recall balance for a class after pruning is proportionally greater in absolute value then before. We will use the following rewording when the manuscript is updated:
>
> *We observe an intensification effect, meaning that for classes with recall below accuracy we observe this gap being negatively widened in proportion to the accuracy after pruning. For classes with recall above accuracy, we conversely observe the gap being positively widened. The intensification correlates with excessive pruning ratios as well as more complex data and models, and is more pronounced with some pruning algorithms.*
>
> We would appreciate any other recommendations on how to reword these or other parts of the paper for better clarity.

---

> ### Author Response · Authors · 2022-08-02
> **Relationship with prior work**
>
> Our work shows that the relationship between class recall and model accuracy is very often affected when neural networks are pruned. Thanks to prior work, it was already known that the recall balance for underrepresented classes and features becomes proportionally more negative with pruning [27,28,52]. In particular, the identification of samples that are affected by pruning in [27,28] has the implicit assumption that changes are negative and should be avoided [7].
>
> In our experiments, we have shown that increased task difficulty, model complexity, and pruning ratios may intensify the effect for classes in which a model already underperforms before network pruning. All of that complements the prior studies with imbalanced datasets. Moreover, we have also observed a relative improvement for classes in which a model overperforms.
>
> More surprisingly, however, is that we have observed the opposite phenomenon at lower pruning ratios. Namely, we observe a de-intensification effect that implies a corrective effect. Consequently, model accuracy may improve at the same time that the differences between class recall and model accuracy go down (see post “Reduction in recall distortion for small amounts of pruning & operationalization”). To the best of our knowledge, no other work has identified this positive externality of pruning on class imbalances.

---

> ### Author Response · Authors · 2022-08-02
> **Motivation and benefit of UP**
>
> The motivation for UP is to address what we believe is a double-edged approach to reducing the number of parameters. Namely, that it is common to use a term in the loss function to make the weights as small as possible, such as weight decay, and then that we use the gradient of the loss function for pruning weights while ignoring that the gradient is also affected by the weight decay term.
>
> It is currently accepted that magnitude-based pruning (MP) works better in practice than gradient-based pruning (GP), even though MP is based on the somewhat misleading assumption that smaller weights have a smaller impact if pruned (counterexamples exist since the 1980s [23,38]), whereas GP is more principled for being based on the effect of pruning the weight in the loss function.
>
> However, if we revisit GP with the premise that the weight decay term should be deducted from the loss function first, then we not only improve the results with GP but we also obtain better results than with MP. Moreover, the way in which the weights are ultimately ranked reflects a weighted combination of the criteria used for MP and GP. This is particularly interesting because UP coincides with MP when the gradients used for GP are sufficiently close to zero, hence providing a principled argument for the effectiveness of MP under weight decay.
>
> Since UP consists of a fine adjustment between the MP and GP approaches, we are not surprised that this leads only to a minor improvement. However, this improvement shows that a better alignment of the pruning method with the loss function alleviates recall balance. We believe that presenting this adjustment as part of the paper illustrates how recall balance issues can be addressed.

---

> ### Author Response · Authors · 2022-08-02
> **Reduction in recall distortion for small amounts of pruning & operationalization**
>
> Regarding your question about the existence of a tradeoff between recall distortion and overall accuracy, we actually observe some agreement between those metrics for the pruning ratios at which both are more beneficial. More specifically, we observe model accuracy improving at the same time that intensification is reduced if a small pruning ratios is used.
>
> We have attached some preliminary plots to the supplementary material (see file AIxR.png), in which we refine our study a lower pruning ratios by evaluating 30 models at ratios 2, 4, 6, 8, and 10. In those plots, the green curve associated with the left y-axis represents the mean accuracy at each compression ratio, with the initial observation at compression ratio 1 corresponding to the model accuracy before pruning. The blue curve associated with the right y-axis represents the mean intensification at each compression ratio. We have aligned both y-axes so that the center of the left axis represents the mean accuracy before pruning and the right axis represents an intensification ratio of 1, and then a black horizontal line is drawn at the center of the plot. Whenever we see the green plot above that line and the blue plot below that line, which is typical for the lower pruning ratios, we are observing accuracy going up while intensification is going down.
>
> Based on those plots, we also agree with your comment that intensification could be another axis along which pruning methods should be evaluated. When the pruning ratio increases, model accuracy and intensification no longer move in the same direction. Whereas heavy pruning makes the accuracy worse and intensification stronger, lighter pruning makes the accuracy better while not pushing  intensification above 1.
>
> We believe that these results are actionable to the extent that they encourage the use of a moderate amount of pruning both for the sake of improving generalization as well as to reduce the performance balance across classes. Moreover, our results provide a qualitative understanding of how to adjust for different cases. Namely, if we work with a comparatively more complex task we should compensate with a larger model or a lower pruning ratio if we would like to obtain a pruned model with similar improvements for both metrics.

---

> > ### Comment · Reviewer_X7NS · 2022-08-06
> > **Preliminary plots**
> >
> > The experiments in AIxR.png do indeed look promising. But it is still a small effect and so I think it would be helpful to see its significance. Can you add standard deviations to these plots?

---

> > > ### Author Response · Authors · 2022-08-06
> > > **Updated plot**
> > >
> > > Thank you.
> > >
> > > We have included another file in the supplementary material, AIxR_2.png, which includes the error bars.

---

> > > > ### Comment · Reviewer_X7NS · 2022-08-07
> > > > **Updated Plot**
> > > >
> > > > Excellent, thank you!

---

### Author Response · Authors · 2022-08-02
**Summarizing our initial response**

We would like to thank the reviewers for the many suggestions and insightful comments about our study. We believe that our responses to each reviewer have addressed most of their concerns, with the only outstanding question being about recall distortion in recent pruning algorithms (last point below). We will include the explanations provided to the reviewers below in the next revision of the paper.

Hence, we would appreciate if the reviewers could please reconsider their assessment of our work based on our responses, or alternatively let us know if they still have concerns that we could try to address within the next days.

In summary, here is in general terms what we addressed (and where we addressed it):

**Clarifications regarding the metrics used** (addressed with reviewer x7NS in post "The metric $\alpha$" and with reviewer imMk in post "Justifying the metrics"):

We described to reviewer imMk that recall is actually a commonly used metric in the literature for the distortions caused by network pruning, although usually presented with other names, and that recall balance is a natural extension if we want to compare recall with accuracy. Normalizing the recall balance is justified by the example that we use in pages 4 and 5. To reviewer X7NS, we explained how calculating alpha leads to an intensification metric that is less affected by outliers than the mean of intensifications for each class.

**Clarifications and suggestions regarding the UP algorithm** (addressed with reviewer X7NS in post "Motivation and benefit of UP" and with reviewer hKwV in post "Accounting for other forms of regularization when pruning"):

We describe the logic and the insights that we obtain from deriving UP to reviewer X7NS, and we acknowledge the interesting suggestions of reviewer hKwV about extending the same idea to implicit normalization techniques. We will look further into that.

**How measuring intensification helps us** (addressed with reviewer X7NS in post "Reduction in recall distortion for small amounts of pruning & operationalization" and with reviewer hKwV in post "How knowing about intensification helps"):

To address both reviewers, we show a more detailed comparison of accuracy and model intensification through a plot included in the Supplementary Material (see figure AIxR.png). This plot shows how it is possible in many cases to use a moderate amount of network pruning to improve generalization while also reducing recall distortion, hence producing a more fair model.

**Significance with respect to the literature** (addressed with reviewer X7NS in post "Relationship with prior work"):

We explain to reviewer X7NS that our work shows that recall distortions may be worsened when using higher pruning ratios, smaller models, and more complex datasets, hence complementing prior studies on the effect of unbalanced datasets. Furthermore, we show for the first time that a moderate amount of pruning may actually reduce recall distortion.

**Minor wording corrections** (addressed with reviewer X7NS in post "Minor points on clarity" and with reviewer imMK in post "Expected results of the experiments"):

We appreciate the comments of the reviewers on parts of the introduction and of the hypothesis testing that could be improved. Any additional suggestions are more than welcome.

**Effect of network pruning on normalized recall balance** (addressed with reviewer imMk in post "Effect of pruning ratio on recall variance"):

Reviewer imMk anticipated that the variance of the normalized recall balance may also be affected as a consequence of intensification. Indeed that is the case, as we have shown in a table in our response.

**Comparison with recent network pruning algorithms** (addressed with reviewer hKwV in post "Comparison with other pruning methods" and with reviewer imMk in post "Comparison with SOTA pruning algorithms "):

We have presented preliminary results on a single model compressed with CHIP (as suggested by reviewer hKwV) and with LTH (to address the comment about SOTA algorithms by reviewer imMk). We believe that is a great addition to the paper to investigate the extent to which intensification has been addressed by more recent approaches. Our preliminary results indicate that intensification may be smaller, but it is still present. We will try our best to update those responses with more experiments in the week of discussion.

While we believe that studying some recent approaches can be very insightful, especially if they have reduced the extent to which recall distortions  occur, we believe that centering our study in the classic algorithms that inspired most of such approaches (many recent papers derive from either MP or GP) would keep our work relevant for a longer period of time. Therefore, we believe that it is actually a strength of the paper that we have focused on classic algorithms that are widely known, in particular if results on SOTA approaches are also reported.

---

### Author Response · Authors · 2022-08-06
**Update: Comparison with more recent methods**

We studied in more detail the effect of the intensification in the recent pruning methods LTH and CHIP on ResNet-56. In both cases, we were able to show that the intensification ratio ultimately increases with the pruning ratio and that an intensification above 1 consistently occurs if the pruning ratio exceeds a certain threshold, which depends on the method. In order to reach that threshold, we have also experiment with higher pruning ratios using those methods. We intend to add this finding to our supplementary material in a future revision of the manuscript.

We also note that it is not straightforward to adapt multiple methods to successfully operate on exactly the same pruning ratios, since there is a lot of engineering in making sophisticated methods work well (such as the amount pruned on each iteration with LTH and the amount pruned from each layer with CHIP). For that reason, we emphasize once more our belief that studying classic methods makes it easier to isolate different factors that may influence intensification, as we did in our study. We appreciate the recommendations by the reviewers to consider other methods, and we believe that including their evaluation as a complement of our current study strengthens the paper while still allowing to compare the many factors that affect intensification.

If the reviewers believe that there are other methods that should be studied, we will work on including them as well.

# Results for LTH

The results below summarize the outcome of 10 runs of LTH (https://arxiv.org/abs/1803.03635) on models trained on the same setting as those of our other experiments using 30,000 steps for training at each level of pruning, with the corresponding pruning ratio next to it. The level 0 (pruning ratio 1) corresponds to the original model without pruning. The LTH paper only goes as far as step 15. We extend the number of steps using the same pruning ratio between steps used up to step 15.

| Level | Pruning ratio | Accuracy | Intensification |
| :---: | :---: | :---: | :---: |
| 0 | 1.00 | 0.855 | _ |
| 1 | 1.25 | 0.854 | 1.010 |
| 2 |  1.56 | 0.859 | 0.986 |
| 3 |  1.95 | 0.859 | 0.970 |
| 4 |  2.44 | 0.859 | 0.974 |
| 5 |  3.05 | 0.859 | 0.970 |
| 6 |  3.81 | 0.859 | 0.932 |
| 7 |  4.77 | 0.858 | 0.957 |
| 8 |  5.96 | 0.856 | 0.973 |
| 9 |  7.45 | 0.853 | 0.982 |
| 10 |  9.31 | 0.850 | 0.980 |
| 11 | 11.64 | 0.846 | 0.997 |
| 12 | 14.55 | 0.841 | 1.029 |
| 13 | 18.19 | 0.835 | 1.055 |
| 14 | 22.74 | 0.826 | 1.131 |
| 15 | 28.42 | 0.811 | 1.244 |
| 16 | 35.53 | 0.795 | 1.293 |
| 17 | 44.41 | 0.769 | 1.377 |
| 18 | 55.51 | 0.739 | 1.536 |
| 19 | 69.39 | 0.694 | 1.767 |
| 20 | 86.74 | 0.669 | 1.910 |

Even if restricted to the first 15 steps, we already observe intensification by step 12. The increase in intensification is consistent since step 6, which is in line with our findings in the paper.

# Results for CHIP

The results below summarize the outcome of 15 runs of CHIP (https://arxiv.org/abs/2110.13981) on models trained on the same setting as those of our other experiments, with the corresponding pruning ratio next to it. Pruning ratio 1 corresponds to the original model without pruning. The CHIP paper only goes as far as pruning ratio 3.33. We extend the number of pruning ratios with ratios 8.27 and 19.11 by preserving the proportion of unpruned weights used for pruning ratio 3.33.

| Pruning ratio | Accuracy | Intensification |
| :---: | :---: | :---: |
| 1.00 | 0.886 | - |
| 1.75 | 0.934 | 0.516 |
| 3.33 | 0.920 | 0.650 |
| 8.27 | 0.882 | 0.985 |
| 19.11 | 0.824 | 1.494 |

If restricted to pruning ratios 1.75 and 3.33, we observe that intensification starts to increase from one pruning ratio to another, which is in line with our findings in the paper.

---

### Meta-Review · Area_Chair_KL8V · 2022-08-29

**Recommendation:** Accept
**Confidence:** Certain

**Metareview:**

This paper studies the disparate effect of model pruning across classes and proposes a new method to reduce the "recall distortion" across classes. This is a critically important problem, and one which has just begun to be carefully studied in the literature, so this work is timely and relevant. All reviewers recognized the relevance of the problem and the novelty of the authors' approach, both with respect to the new approach presented here, as well as the detailed analysis of the various factors which impact recall distortion. There were some concerns regarding the complexity of the pruning algorithms studied, but the authors provided a number of additional experiments on other pruning approaches in their response, finding qualitatively similar effects (as might be expected given the reliance of many of these approaches on some form of magnitude pruning). I think this paper will be a valuable addition to a poorly understood and important research area, and should be accepted.

**Award:**

No

---

### Decision · Program_Chairs · 2022-09-14

Accept